# Compression via Pre-trained Transformers:
# A Study on Byte-Level Multimodal Data

David Heurtel-Depeiges [* † 1]   Anian Ruoss [* 2]   Joel Veness [2]   Tim Genewein [2]

## Abstract

Foundation models are strong data compressors, but when accounting for their parameter size, their compression ratios are inferior to standard compression algorithms. Naively reducing the parameter count does not necessarily help as it deteriorates predictions and, accordingly, compression. We conduct a large-scale empirical study to find a sweet spot where pre-trained vanilla transformers can achieve competitive compression ratios. To this end, we train models on 165GB of raw byte sequences of either text, image, or audio data (and all possible combinations of the three) and then compress 1GB of out-of-distribution (OOD) data from each modality. We find that relatively small models (millions of parameters) can outperform standard general-purpose compression algorithms (gzip, LZMA2) and even domain-specific compressors (PNG, JPEG-XL, FLAC) — even when accounting for parameter size. We achieve, e.g., the lowest compression ratio of $0.49$ on OOD audio data (vs. $0.54$ for FLAC). We conduct extensive ablations and hyperparameter sweeps to study the impact of model- and dataset scale, and we investigate the effect of unimodal versus multimodal training. We find that even small models can be trained to perform well on multiple modalities, but unlike large-scale foundation models, transfer to unseen modalities is generally weak.

## 1. Introduction

Strong predictive models can straightforwardly be turned into strong lossless compressors, e.g., via arithmetic coding (Pasco, 1977; Rissanen, 1976; Witten et al., 1987). Consequently, large pre-trained foundation models, such as LLMs, achieve high data compression on their training distributions and beyond (Delétang et al., 2024). However, when factoring these models' size into the compression ratio, too large models actually perform worse. For this reason, large foundation models with billions of parameters cannot compete with standard compression algorithms such as gzip (Deutsch, 1996) or LZMA2 (Pavlov, 2019). The goal of this paper is thus to investigate whether pre-trained vanilla transformers can achieve compression ratios that are competitive with standard algorithms across a range of data modalities. This places fairly tight constraints on the maximal model size, leading us to investigate families of relatively small transformers (with millions of parameters). Note that our aim is not to build a practical transformer-based data compressor, as the computational footprint (running time, memory, FLOPs) of even small models is far beyond standard compressors. Instead, studying compression via pre-trained models provides insight into the models' *learned* inductive biases, e.g., whether they are domain-general, how they depend on the training data composition, and whether there is transfer between modalities.

Recently, Delétang et al. (2024) stated that "language modeling is compression", pointing out that log-loss minimization is equivalent to optimizing a lossless compression objective. To illustrate this point, Delétang et al. (2024) used billion-parameter LLMs that were exclusively trained on text (Touvron et al., 2023b; Hoffmann et al., 2022) to compress 1GB of ImageNet images (Russakovsky et al., 2015) and LibriSpeech audio (Panayotov et al., 2015), respectively. They found that these models compress better than gzip or LZMA2 and even domain-specific compressors such as PNG (Boutell, 1997) or FLAC (Coalson, 2008), but only when parameter counts are not considered. However, when evaluating *pre-trained* neural networks as compressors, the parameter count has to be factored into the compression ratio. To illustrate this, imagine that Alice wants to compress and send data to Bob, who wants to decompress it. If Alice trains a neural network to compress the data but does not communicate the model's weights, Bob cannot decode the data (unless Alice communicates the training recipe and Bob retrains the same network from scratch, but that would no

---

[*]Equal contribution [†]Work performed while the author was at Google DeepMind [1]Chandar Research Lab. MILA - Quebec AI Institute Polytechnique Montréal [2]Google DeepMind. Correspondence to: Anian Ruoss <anianr@google.com>, Tim Genewein <timgen@google.com>.

*Proceedings of the $42^{nd}$ International Conference on Machine Learning*, Vancouver, Canada. PMLR 267, 2025. Copyright 2025 by the author(s).

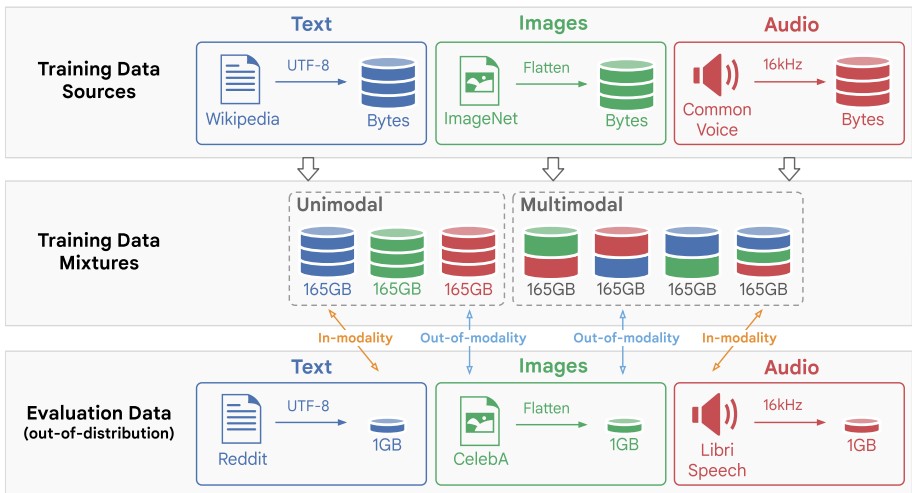

Figure 1: Our training and evaluation data pipelines. We consider three modalities: text, images, and audio. From these we create training mixtures of 165GB that are either unimodal or multimodal. After pre-training transformers on each of these, we evaluate their compression ratio (factoring in the model size) on all three modalities. If the corresponding modality has not been seen during training, the evaluation is 'out-of-modality', otherwise it is 'in-modality'. Importantly, our evaluation is always performed on out-of-distribution data (different from any of the training data sources), even when it is in-modality.

longer fall into the pre-trained regime). Accordingly, when evaluating pre-trained neural networks as compressors, the parameter count needs to be factored into the compression ratio. For this reason Delétang et al. (2024) also trained small-scale transformers (up to 3.2M parameters) on 1GB of Wikipedia (Hutter, 2006) but found that these models were significantly worse at compressing images and audio.

The obvious *open question* (see Appendix A) is whether small transformers pre-trained on large (multimodal) datasets can achieve competitive compression ratios across different modalities and whether there is transfer to unseen modalities, as observed in the large-scale model case. We therefore conduct an extensive empirical study where we train families of decoder-only transformers on 165GB of either text, image, or audio data and all combinations of the three. We then use these models (with frozen parameters, i.e., offline training) to compress 1GB of out-of-distribution (OOD) data from all three modalities (see Fig. 1). We compare against transformers trained purely online, i.e., on the data stream that is being compressed (Bellard, 2019; 2021; Izacard et al., 2020), for which storage/communication of the weights is not required for decompression (unlike our pre-trained models). These online transformers are currently state-of-the-art on the Large Text Compression Benchmark (Mahoney, 2006). Overall we find that our small pre-trained transformers achieve competitive compression ratios, consistently outperform domain-general and domain-specific standard compression algorithms, and are on par with the online transformers from Bellard (2021).

**Main Contributions** Our key contributions are:

- We conduct a large-scale empirical study (hyperparameter sweeps, ablations) on the compression performance of small transformers pre-trained on raw byte sequences of text, image, and audio data (and all combinations), across various model- and dataset sizes.

- We are the first to show that *small pre-trained* transformers achieve better compression ratios than general-purpose and domain-specific compressors on 1GB of OOD data across different modalities, e.g., 0.49 on audio vs. 0.51 for Bellard (2021) & 0.54 for FLAC.

- We show that training on multiple modalities only slightly deteriorates the performance on each individual modality but significantly boosts the compression ratios on multimodal data, as long as all the evaluation modalities are part of the pre-training data mixture.

- We show that small pre-trained transformers fail to beat standard compressors on data modalities that were unseen during training, implying weak out-of-modality transfer (unlike LLMs, see Delétang et al. (2024)).

## 2. Background

Compression and prediction are "two sides of the same coin" (MacKay, 2003). This fundamental duality stems directly from Shannon's celebrated lossless source coding theorem (Shannon, 1948), which states that there is a well-defined lower bound for encoding data from a probabilistic

source. For any data sequence $x_{1:n} := x_1 x_2 \ldots x_n \in \mathcal{X}^n$ of length $n$ from a finite alphabet $\mathcal{X}$ sampled from a source $\rho : \mathcal{X}^* \mapsto (0, 1]$, a lossless compressor $c : \mathcal{X}^* \mapsto \{0, 1\}^*$ assigns a code $c(x_{1:n})$, i.e., a sequence of bits, from which the original sequence is recoverable without loss of information. The goal is to minimize the expected length $L_\rho := E_{x \sim \rho}[\ell_c(x)]$ by encoding rare sequences with more bits and frequent sequences with fewer bits. Shannon's source coding theorem states that the minimal expected length is lower-bounded by the Shannon entropy of the source: $L_\rho \geq H(\rho) := \mathbb{E}_{x \sim \rho}[-\log_2 \rho(x)]$.

If the source's statistics are unknown, good compression becomes a statistical modeling problem, i.e., it relies entirely on being able to predict well sequentially. For any predictor $\pi : \mathcal{X}^* \mapsto (0, 1]$ the expected coding length $L_\pi^\rho$ for data drawn from $\rho$ is at least the cross entropy:

$$L_\pi^\rho \geq \mathbb{E}_{x \sim \rho}[-\log_2 \pi(x)] = \mathbb{E}_{x \sim \rho}\left[-\log_2 \frac{\pi(x)\rho(x)}{\rho(x)}\right]$$
$$= H(\rho) + D_{\mathrm{KL}}(\rho || \pi) \geq H(\rho),$$

which is also lower-bounded by the Shannon entropy of $\rho$. A mismatch between $\pi$ and $\rho$ thus leads to an excess length given by their KL divergence, and minimal coding length (maximal compression) implies $\pi = \rho$ across the whole support of $\rho$. Accordingly, some AI researchers have argued that compressing well is fundamentally connected to intelligence (e.g., Chaitin's famous "Compression is Comprehension" (Chaitin, 2006); Rathmanner & Hutter (2011); Grau-Moya et al. (2024)), and that building universal compressors will accelerate AI development (cf. the Hutter prize (Hutter, 2006), an ongoing competition to compress (1GB of) human knowledge). The duality between compression and prediction has also led to the (algorithmic) information-theoretic formulation of universal prediction, i.e., Solomonoff induction (Solomonoff, 1964a;b; Li & Vitányi, 2019), one of two key ingredients for AIXI (Legg & Hutter, 2007; Hutter et al., 2024), the theory of artificial superintelligence.

Consequently, Delétang et al. (2024) argue that lossless compression performance lends itself as a domain-general metric for assessing any predictor's quality, including foundation models. They further emphasize that foundation models trained by minimizing log-loss (a.k.a., next-token prediction-error or cross entropy loss) are explicitly trained to minimize the expected coding length:

$$\min_\pi L_\pi^\rho = \min_\pi \underbrace{\mathbb{E}_{x \sim \rho}[-\log_2 \pi(x)]}_{\text{"log loss"}} \tag{1}$$

$$= \min_\pi \mathbb{E}_{x \sim \rho}\left[\sum_i -\log_2 \pi(x_i | x_{<i})\right] \tag{2}$$

The problem of constructing the actual codes that achieve (near) minimal expected length given a predictor is largely solved in information theory, with gold-standard algorithms such as Huffman coding (Huffman, 1952), arithmetic coding (Pasco, 1977; Rissanen, 1976; Witten et al., 1987), or asymmetric numeral systems (Duda, 2009). The latter two compress strings online by iteratively converting them into a single binary number with increasing precision (see Delétang et al. (2024) for an illustration or Chapter 2 in Hutter et al. (2024)). Arithmetic coding is an example of an online compression algorithm since it only requires a single pass through the data and compresses on the fly (unlike offline compressors, such as Huffman coding). Both our models and Bellard (2021), which we compare against, use arithmetic coding and compress online. However, the difference is that we pre-train our predictor, i.e., we perform *offline training* on a dataset and then freeze its parameters (non-adaptive arithmetic coding), whereas Bellard (2019; 2021) and Izacard et al. (2020) perform *online adaptation* of the model parameters on the data stream that is being compressed (adaptive arithmetic coding). As a result, and unlike our compressors, these approaches do not communicate the trained weights for decompression but only the model architecture and training algorithm (i.e., the compression ratio does not need to account for the model parameters).

## 3. Related Work

**Compression Without Transformers**    Lossless compression with (non-transformer) neural predictors has been extensively studied, both via arithmetic coding (Lehtokangas et al., 1993; Schmidhuber & Heil, 1994; 1996; Mahoney, 2000; Mikolov, 2012; Knoll, 2014; van den Oord & Schrauwen, 2014; Cox, 2016; Schiopu et al., 2018; Goyal et al., 2019; Liu et al., 2019; Mentzer et al., 2019; 2020; Schiopu & Munteanu, 2020; Rhee et al., 2022) and asymmetric numeral systems (Hoogeboom et al., 2019; Kingma et al., 2019; Townsend et al., 2019; Barzen et al., 2022). Neural networks are also used for lossy compression, e.g., by overfitting tiny networks to data points and transmitting the weights rather than the data (Dupont et al., 2021; 2022; Chen et al., 2021; Ladune et al., 2023; Kim et al., 2024).

**Online Transformers**    Most of the above approaches use a separate training set to pre-train models that are then used to compress a test set. Alternatively, the model can also be trained from scratch on the data stream that is being compressed (Bellard, 2019; 2021; Izacard et al., 2020; Goyal et al., 2020; Mao et al., 2022). The main advantage of these adaptive online compressors is that they are (quasi) parameterless (since they are initialized from scratch when compressing a data stream), i.e., the model size does not explicitly affect the compression ratio, even for large models (though it implicitly affects the training performance, e.g., large models train more slowly meaning that larger chunks of the data stream are only weakly compressed).

The transformer-based adaptive online compressor of Bellard (2021) is currently state-of-the-art on the Large Text Compression Benchmark (Mahoney, 2006), and Section 5 shows that our best models are on par across all modalities.

**Pre-Trained Transformers** Most related to our work is the line of research by Valmeekam et al. (2023); Delétang et al. (2024); Huang et al. (2024); Li et al. (2024); Mittu et al. (2024); Chen et al. (2024) on lossless compression via arithmetic coding with pre-trained foundation models, i.e., the Llama models (Touvron et al., 2023a;b; Dubey et al., 2024) and Chinchilla (Hoffmann et al., 2022). Delétang et al. (2024), in particular, also report good compression rates on unseen modalities (LLMs trained only on text compress images and audio data well). However, these studies differ from our work as they do not take the model size into account for the compression ratios, except for Delétang et al. (2024), who report both "raw" and "adjusted" compression ratios and find that LLMs are not competitive in terms of adjusted (i.e., the actual) compression ratios. To the best of our knowledge, our paper is the first to systematically investigate the use of *appropriately-sized pre-trained transformers* for multimodal lossless compression in a regime where competitive performance w.r.t. standard compression algorithms is possible. In this regime, our study is the most comprehensive in that it also investigates multimodal training and cross-modal transfer of pre-trained transformers.

## 4. Methods

We now describe our experimental setup (cf. Appendix B).

**Baselines** We compare to various standard compressors, both general-purpose, i.e., gzip (Deutsch, 1996) and LZMA2 (Pavlov, 2019), and domain-specific, i.e., FLAC (Coalson, 2008) for audio data and PNG (Boutell, 1997) and lossless JPEG 2000 (Skodras et al., 2001) and JPEG-XL (Alakuijala et al., 2019) for images. Both gzip and LZMA2 (which is used by the 7zip software) are based on Huffman coding (Huffman, 1952) and the Lempel-Ziv-Welch algorithm (Welch, 1984). We use the default parameters for gzip, LZMA2, JPEG 2000, and JPEG-XL, compression level 12 for FLAC, and instruct PNG to find the optimal encoder settings. We also compare to the online transformer from Bellard (2021) with the default v3.3 parameters, which is currently state-of-the-art on the Large Text Compression Benchmark (LTCB) (Mahoney, 2006). We do not compare to Izacard et al. (2020), since it is conceptually very close to Bellard (2021) but worse on LTCB.

**Models** We focus on decoder-only transformers (Vaswani et al., 2017) with SwiGLU activations (Shazeer, 2020) and post-layer normalization. Unless otherwise noted, we use 8 heads, an embedding dimension of 64, a context size

of 4096 (bytes), and sliding windows without overlap or memory (full details in Appendix B.3). We always train and evaluate the models with the same context size (4096 by default). We use the Adam optimizer (Kingma & Ba, 2015) for 2.5 million steps with a batch size of 32, which, for 165GB of data, roughly corresponds to 2 epochs. Due to the duality of compression and prediction, we minimize the standard (sequential) log-loss (Eq. (1)) during training, which is a maximum-compression objective (see Section 2). The model computes a distribution over the next byte for every input byte, and the arithmetic coder then uses these predictions to losslessly compress the data. Concretely, the arithmetic coder directly uses the model's predictions over tokens (i.e., the logits) to encode/decode data (see Figure 1 in Delétang et al. (2024) for an overview). Accordingly, no separate head or extraction procedure is necessary. As a result, we can train via standard log-loss minimization to perform next-byte prediction. At inference time, we perform standard autoregressive evaluation (using teacher forcing).

**(No) Tokenization** Tokenization is a commonly-used, *domain-specific* pre-compression step to boost transformers' performance by increasing their vocabulary size to fit more information into their context window (Lester et al., 2024), i.e., increased information density at the cost of increased entropy. However, since we aim to be domain-general, we do not use tokenization and instead feed byte streams to our models (we still have to choose how to flatten images/sample audio signals, i.e., minimal domain-specific preprocessing).

**Evaluation** To evaluate performance, we compute the compression ratio (lower is better):

$$\text{compression ratio} := \frac{|\text{compressed data}| + |\text{compressor}|}{|\text{uncompressed data}|}, \tag{3}$$

which accounts for the model size and is equivalent to the "adjusted compression rate" of Delétang et al. (2024). We always evaluate on 1GB of out-of-distribution data, i.e., $|\text{uncompressed data}| = 1\text{GB}$. Like Delétang et al. (2024), we compute the size of the compressor by encoding the model weights with `float16` (2 bytes per parameter) since this level of quantization has little impact on performance (Tao et al., 2022) and is standard for model inference. As a result, our model sizes range from 0.8MB to 40.3MB. Note that, similar to Delétang et al. (2024), we do not compress the model parameters, since naive approaches (e.g., compressing them with gzip) barely decrease the model size (only by around 7%, which corresponds to a decrease in compression ratio of only 0.002821 for our largest model). However, as a result, the compression ratio we report is an upper bound, which could be improved by (losslessly) compressing the parameters (though with limited room for improvement in our regime, even in the best case).

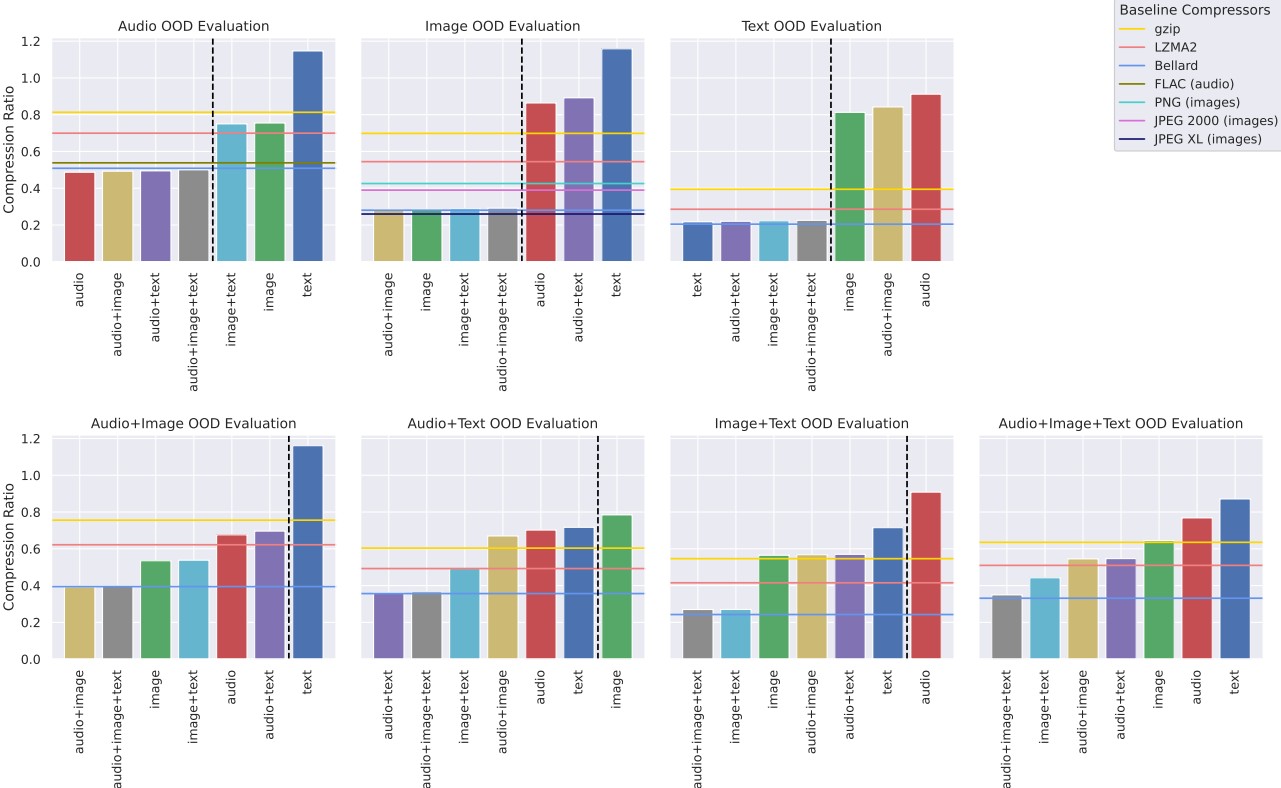

Figure 2: Small pre-trained transformers are domain-general compressors (panels: evaluation data mixtures, bars: training data mixtures). Our method (bars) outperforms standard compression algorithms (horizontal lines) and is on par with the online adaptive transformers from Bellard (2021) (blue line) — as long as the evaluation modality is in the training mixture. There is very little cross-modal transfer to unseen modalities (unlike foundation models (Delétang et al., 2024)). Unimodal models are good for their respective modality, but multimodal models perform almost as well across all their training modalities (despite seeing much less data per modality than the unimodal models), i.e., one can trade off a small amount of performance on each individual modality to obtain a strong domain-general compressor via multimodal training (gray bar).

**Training Datasets** A key point of our investigation is to evaluate how well pre-trained transformers can compress data from *different modalities* — both if the modality was or was not part of the training data (see Fig. 1). We create three different unimodal training datasets with audio, images, and text, and four multimodal training sets (full details in Appendix B.1). This yields seven 165GB pre-training datasets: (i) 165GB audio; (ii) 165GB images; (iii) 165GB text; (iv) 82.5GB audio and 82.5GB images; (v) 82.5GB audio and 82.5GB text; (vi) 82.5GB images and 82.5GB text; and (vii) 55GB audio, 55GB images, and 55GB text. By training models on all seven data mixtures, we can investigate *in-modality* and *out-of-modality* compression ratios. For example, for a model trained on text (iii), the in-modality compression ratio is given by evaluating on text, while audio or image data provide out-of-modality compression ratios.

**OOD Evaluation Datasets** To mimic the standard compression algorithm setting (and thereby ensure a fair com-

parison), where the compressor is designed with only minimal statistical assumptions about the data (domain-specific compressors make stronger assumptions), we evaluate on unseen, out-of-distribution (OOD) datasets for each modality and not on in-distribution held-out datasets (as common in machine learning). To do so, we create a single OOD dataset of 1GB for each modality (details in Appendix B.2).

## 5. Results

We present our extensive experimental evaluation (see also Appendix C). Unless otherwise noted, we report the best results over two hyperparameter sweeps (see Appendix B.3): (i) model- vs. dataset size, and (ii) model- vs. context size.

**Small Transformers Are Domain-General Compressors** Figure 2 shows the best compression ratio attained on each of the seven out-of-distribution evaluation datasets when training a model on each of the seven training data mix-

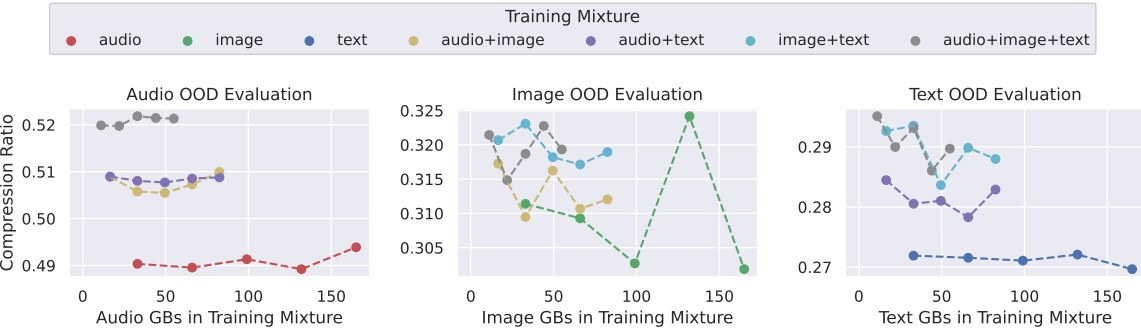

Figure 3: What you see is what you get. Each panel visualizes the compression ratios for one of our modalities when training models on varying dataset mixtures and sizes. Although one can replace a large proportion of the unimodal training datasets with data from other modalities without incurring significant losses on the original modality (note the scale of the y-axis), transformers (at our tested model sizes) do not exhibit improved transfer from the out-of-modality data (i.e., the multimodal models are worse than the unimodal ones, even when trained on much more data from that particular modality). Nevertheless, multimodal training data significantly improves multimodal compression performance (as shown in Fig. 2).

tures (we report the best-performing model from our two sweeps for each training-evaluation pair). We observe that pre-trained transformers achieve state-of-the-art in-modality compression ratios, regardless of the concrete composition of the training mixture, outperforming standard compression algorithms (even domain-specific ones) in all cases where all evaluation modalities are part of the training mixture. In these cases, transformers thus learn the modality's prototypical statistical patterns during pre-training. Importantly, by comparing models trained on unimodal vs. multimodal data, we observe that multimodal training only slightly decreases performance compared to the unimodal models on their respective modalities (despite only having half or a third amount of data from that modality). This means that it is possible to trade off a small amount of performance on each individual modality to obtain a very strong domain-general compressor via multimodal training (the gray bar in Fig. 2).

**What You See Is What You Get**  While Fig. 2 shows that substituting half or two thirds of the training set with data from other modalities only leads to a small performance loss compared to the unimodally trained models, it is unclear whether simply training on a smaller amount of unimodal data (i.e., decreasing the unimodal training dataset size to, e.g., 82.5GB, and not substituting 82.5GB with another modality) would give the same performance, or whether there is some transfer between modalities (as suggested by Mirchandani et al. (2023)) that compensates for the smaller amount of data per individual modality. To investigate this, we run an ablation where we subdivide each of our seven training sets into 5 different sizes: 20%, 40%, 60%, 80%, and 100% of the respective dataset (uniformly subsampled). We train a series of models (sweeping over their number of layers; see Appendix B.3) on each dataset mixture and each

dataset size, and then evaluate as before. Figure 3 shows that, for our models and datasets, there is little transfer between modalities. For all cases of audio, text, and (less clearly) images, it is better to train on a smaller unimodal dataset to get the best unimodal performance, as opposed to training on a much larger multimodal dataset. For example, training on a pure text dataset of 33GB (20% of 165GB) outperforms training on a dataset consisting of 82.5GB (i.e., more than twice as much) text and of 82.5GB images/audio.

**Scaling Analysis**  Since there is a non-trivial relationship between model- and dataset size, we analyze the scaling of these factors (details in Appendix B.3). Figure 4 shows trends akin to the scaling laws observed for LLMs (Kaplan et al., 2020), which state that better prediction (in our case compression) is only possible by scaling both models and datasets in a particular way. Unlike traditional scaling laws for models trained on internet-scale datasets, the distribution shift in our evaluation makes it easier for the model to overfit to the training distribution (since we always evaluate on OOD data). However, as the number of parameters and the training flops of our small models increase, the adjusted compression ratio improves, eventually beating standard compression algorithms. We do observe gradual overfitting on the image dataset for our models trained only on images. However, this phenomenon can be mitigated by including other modalities in the training mixture (see Fig. A1).

**Model Size vs. Context Size**  The previous two experiments investigated the impact of training dataset- and model size, revealing a complex, "scaling law"-like, relationship between the two and the overall FLOPS training budget. We now investigate the impact of the context window length. Since the context window length has a large impact on the

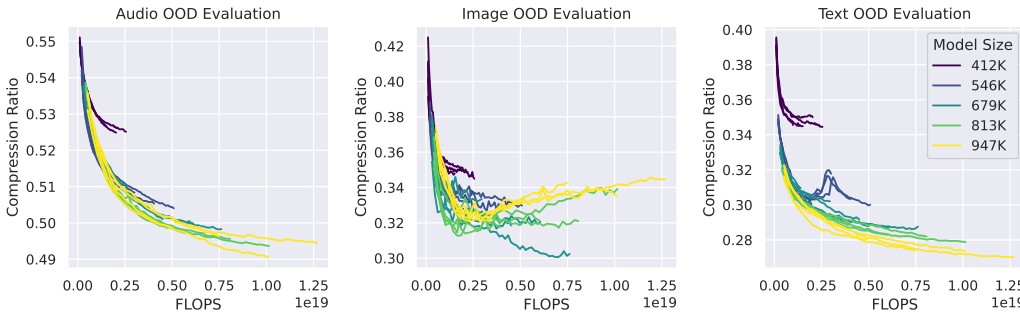

Figure 4: Scaling training dataset- and model size (for unimodal training and evaluation). Colors indicate the model size; lines correspond to dataset size. We train for 2 epochs regardless of dataset size (i.e., smaller datasets require fewer FLOPS). Increasing the model- and dataset size boosts compression (at the cost of FLOPS). Our OOD evaluation makes models more prone to overfitting (e.g., our largest image models), making scaling more complex than traditional LLM scaling laws.

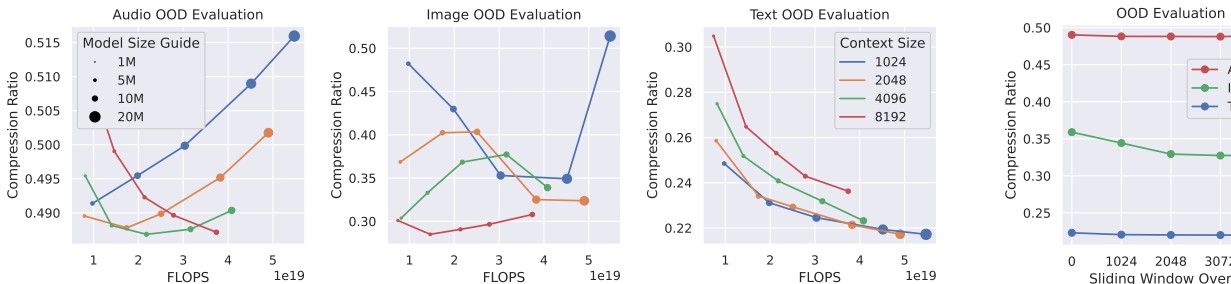

Figure 5: Context- vs. model size. Both context size (measure in bytes) and model sizes affect the training compute budget (in FLOPS), leading to a non-trivial trade-off. Our results show that this trade-off is highly modality-dependent (note the different y-axis scales, i.e., the effect varies significantly with modality). For text, shorter context sizes and larger models are beneficial (short-term dependencies are most important). For images, larger context is generally beneficial, given that a single image consists of $512 \cdot 512 \cdot 3 = 786432$ bytes, far exceeding our models' contexts, i.e., models with larger context can process larger fractions of an image at once. For audio, the relationship is complex with intermediate context length and larger models performing better (the reverse is true for short contexts).

Figure 6: Impact of the sliding window overlap (for uni-modal training/evaluation). Increasing the overlap between context windows only marginally improves the performance (most significantly for images) but comes at a huge computational cost.

overall FLOPS footprint (attention scales quadratically with input sequence length), we also vary the model size to explore whether there is a sweet spot in terms of training compute budget allocation (details in Appendix B.3). Fig. 5 shows that the optimal trade-off strongly depends on the data modality. The best models for text have a context window $\leq 2048$ bytes, indicating that short term dependencies are more important than long ones in this case. For images, the best compromise overall is to choose a larger context window of $8192$, which means decreasing the model size. For audio data, the trade-off is even more complex. Overall these results highlight the difficulty of tuning architectures to achieve best performance across many modalities.

**Sliding Window** So far we used a sliding window without overlap to process the evaluation data. Consequently, bytes

early in the context window are not conditioned on a lot of data (conditioning on more data should help with prediction and thus compression, according to transformers' in-context learning abilities (Brown et al., 2020; Genewein et al., 2023; Ge et al., 2024)). Sliding the context window with more overlap requires more forward passes to process the same amount of data, significantly increasing the computational cost. Figure 6 shows that increasing the overlap window (for a context length of $4096$) has relatively little effect. The strongest effect is observed for image data, since $4096$ bytes only captures a small fraction of an image and there are obvious long-range dependencies between image channels.

**Evaluation Dataset Size** Figure 7 shows the relationship between the compression ratio and the evaluation dataset size for all three modalities and our best-performing model

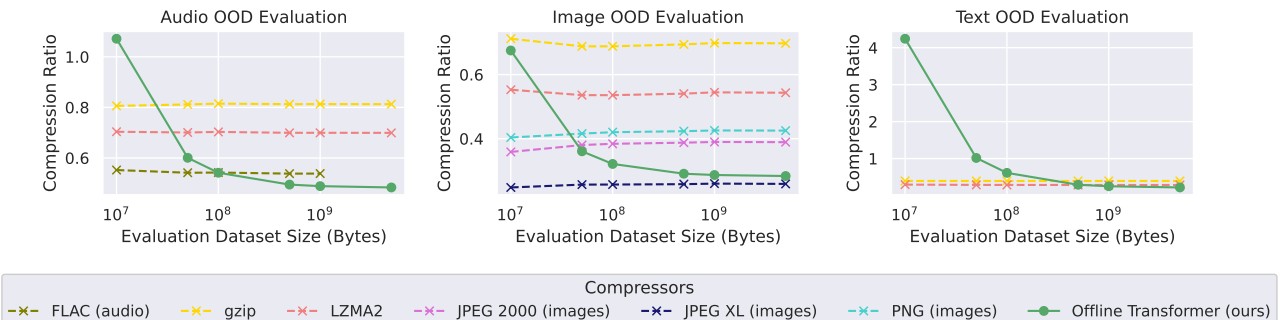

Figure 7: Compression ratio vs. evaluation dataset size. The numerator of the compression ratio consists of the size of the compressed data *and* the size of the compressor (Eq. (3)). For standard compressors (e.g., gzip), the size of the compressor (a few thousand lines of code) is negligible given sufficient evaluation data (i.e., the compression ratio is unaffected by the evaluation dataset size). For neural compressors trained offline (i.e., where the size of the compressor is dominated by the model parameters), the compression ratio improves with increasing data since the model size influence decreases. If the model size $\geq$ evaluation dataset size (e.g., 500M parameters and 1GB of data), one cannot achieve a compression ratio $< 1$.

(as determined on 1GB of OOD data in Table A2). For offline (i.e., pre-) trained neural compressors, the model parameters are factored into the compression ratio, which means that their compression performance will improve with increasing evaluation data (as long as the model generalizes well to the additional data). In contrast, the size of standard compressors is negligible compared to the amount of evaluation data, which means that their compression ratios are largely unaffected by the evaluation dataset size. FLAC cannot losslessly compress more than 4.2GB of data.

## 6. Discussion

Our main goal is to investigate whether pre-trained transformers can compete with standard compressors, even when taking their parameter size into account. In contrast to previous work, this places our models into a relatively small regime, where it is unclear whether models will learn well from large datasets at all and have non-trivial out-of-distribution and cross-modality transfer. One could try to train larger models and compress the model parameters themselves, but we chose not to do this since naive lossless compression of model parameters leads to $\leq 10\%$ reduction (see Table A4), and even best-case scenarios would only lead to marginal improvements in compression ratio given the size of our models. For very large (e.g., foundation) models, compressing weights to achieve competitive compression ratios may be interesting, though it would require lossy compression (Tao et al., 2022), leading to non-trivial trade-offs between high (lossy) compression and maintaining strong predictor performance (the two summands in the numerator of Eq. (3)). Exploring these trade-offs is an interesting direction for future research but beyond the scope of our work. Another option for larger models is to evaluate on

more test data. We chose 1GB of test data as a regime where standard compression algorithms are hard to beat. Moreover, evaluations on more test data, and/or without accounting for model parameters, have been conducted (Delétang et al., 2024; Valmeekam et al., 2023; Li et al., 2024) (finding significant cross-domain transfer, unlike our experiments).

Like Xue et al. (2022), we do not use a tokenizer, which has two reasons. First, tokenizers are typically pre-trained per modality, and we want to rule out bad cross-modality transfer resulting from a bad tokenizer. Second, tokenization acts as a "pre-compression" step (cf. Delétang et al. (2024)). This pre-compression increases information density in the context window at the cost of increasing entropy, which can make the prediction problem harder: Lester et al. (2024) even show that when using a strong neural-based pre-compressor (together with arithmetic coding) to train LLMs, training performance can collapse catastrophically.

**Limitations**  All our claims regarding the universality of our compressors (or the lack thereof) are limited to the model size, regime, and the particular modalities and datasets we studied. We cannot rule out that there are cases where even in-modality transfer is weak (e.g., using another OOD image evaluation dataset with very different statistics), or that there are cases of non-trivial cross-modal transfer (which we have not observed). We did not investigate transfer learning approaches to improve the out-of-modality performance of our neural compressors, but we consider this an interesting avenue for future work. Our claims regarding outperforming standard compression algorithms are limited to our experiments. We cannot rule out that there are datasets where no pre-trained transformer outperforms, e.g., LZMA2 (in fact, we think its plausible that such datasets can be constructed synthetically). Moreover, we cannot rule out

that more sophisticated architectures (e.g., Perceivers (Jaegle et al., 2021), MegaByte (Yu et al., 2023), or Byte Latent Transformers (Pagnoni et al., 2024)), would outperform our models, and we consider neural architecture optimization for lossless compression an interesting direction for future research. Finally, our goal is not to build practical transformer-based universal compressors to compete with standard compressors in terms of computational footprint. As Table A3 shows, our models are orders of magnitude slower for encoding data (with significantly larger memory- and FLOPS-demands) and $\approx 3$x slower than Bellard's online adaptive transformer. When decoding data with our models, which has to be done token-by-token to obtain the correct conditioning, our running time is even worse.

**Future Work**    Even our largest models that were trained on multiple different modalities failed to achieve out-of-modality transfer (unlike LLMs, which do achieve cross-modal transfer when used as compressors, albeit using the "unajdusted" compression ratio; see Delétang et al. (2024)). Accordingly, investigating more sophisticated training strategies to improve cross-modal transfer, e.g., transfer learning, presents an interesting direction for future work.

## 7. Conclusion

We showed that pre-trained vanilla transformers are competitive "zero-shot" compressors on OOD evaluation data, achieving better compression ratios than domain-general and domain-specific standard compression algorithms. We showed this for text, images, and audio data, and for all possible combinations of the three — as long as the corresponding modalities are in the training mixture. Despite their relatively small size, our models compress well across multiple modalities, without losing much performance compared to a purely unimodal model. On the other hand, multimodal training does not lead to strong compression performance on unseen modalities. This is in contrast to LLMs (Delétang et al., 2024), indicating a qualitative difference between small and (very) large models, even when the small models are trained on large datasets. Our results suggest that small transformers can be pre-trained to exploit statistical regularities on par or better than standard compressors and current state-of-the-art adaptive online neural compressors, but we do not observe the emergence of universal compression.

## Impact Statement

This paper presents work whose goal is to advance the field of machine learning. There are many potential societal consequences of our work, none which we feel must be specifically highlighted here.

## Acknowledgments

We thank Anna Mitenkova, Benjamin Beyret, Emilien Dupont, Daniele Calandriello, Grégoire Delétang, Jordi Grau-Moya, Li Kevin Wenliang, Laurent Orseau, Marcus Hutter, Matthew Aitchison, Sarath Chandar, Satinder Baveja, Theophane Weber, Zhengdong Wang, Zoubin Ghahramani, and the anonymous reviewers for their helpful feedback and insightful discussions.

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

Table A1: Similarities and differences of our work w.r.t. Bellard (2021) and Delétang et al. (2024). Unlike Bellard (2021), we use offline (i.e., pre-) training and consider multimodal data. Online training imposes different constraints on the optimal model size, since the compression ratio does not have to account for the model parameters, which is why Bellard (2021) were able to study somewhat larger models. We also conduct comprehensive scientific ablations over dataset-, model-, context-size, etc. Unlike Delétang et al. (2024), we (only) consider the adjusted (i.e., the actual) compression ratio, which includes the model parameters, and therefore, we cannot use massive foundation models. Moreover, while Delétang et al. (2024) studied the cross-domain transfer of foundation models trained on text, they did not study multimodal training.

| | | Bellard (2021) | Delétang et al. (2024) | Ours |
|---|---|---|---|---|
| | # of Parameters | 187M | 70B | 20M |
| | Training | online | offline | offline |
| | Multimodal | - | evaluation | training & evaluation |
| | Actual (Adj.) Compression Ratio | ✓ | ✗ | ✓ |
| | Model Size | | ✓ | ✓ |
| | Context Size | | ✓ | ✓ |
| | Tokenizer | | ✓ | |
| Ablations | Data Mixture | | | ✓ |
| | Train Dataset Size | | | ✓ |
| | Evaluation Dataset Size | | | ✓ |
| | Sliding Window Overlap | | | ✓ |

# A. Relation to Prior Work

As stated in Section 1 and shown in Table A1, we study the *open question* of whether small transformers pre-trained on multimodal data can achieve competitive compression ratios across different modalities and whether there is transfer to unseen modalities (as observed in the large-scale model case). Consequently, the fact that they can do so cannot be predicted from the strong compression performance of LLMs (e.g., results by Delétang et al. (2024)).

In contrast to Bellard (2021), we use offline adaptive arithmetic coding, which induces entirely different constraints on the optimal model size (since the model parameters do not have to be factored into the compression ratio). Bellard (2021) relies on "test-time" gradient updates on in-distribution data, whereas we pre-train models and then leverage in-context learning on out-of-distribution data. It is, therefore, impossible to conclude from the results in Bellard (2021) whether our results are possible. The training data and protocol are incomparable, making it all the more interesting that our results are generally very close — another surprising finding that cannot be trivially explained.

In contrast to Delétang et al. (2024), we do not evaluate off-the-shelf, large-scale, text-based foundation models but pre-train small-scale transformers on audio, image, and text data. As a result, the compressors proposed by Delétang et al. (2024) are not competitive w.r.t. standard compressors (which they acknowledge) — unlike our models (we beat standard compressors across the board). Delétang et al. (2024) do conduct a pilot experiment by pre-training a small transformer on text data, but, in contrast to our work, they do not perform (i) a comprehensive study of multi-modal training, (ii) out-of-distribution evaluation, (iii) ablations over context-, dataset-, and model-size, (iv) a comparison to Bellard (2021), (v) an investigation of the sliding window overlap, and (vi) a test dataset size ablation. As our work shows, all of these ingredients are necessary to obtain the strong performance we reported, and, as a result, Delétang et al. (2024) do not manage to pre-train a transformer-based compressor that beats standard compressors across multiple modalities — unlike our work. The main contribution of Delétang et al. (2024) is conceptual, with a relatively simple experimental evaluation to illustrate the conceptual arguments. Our work, on the other hand, performs a rigorous and comprehensive empirical study.

# B. Experimental Details

### B.1. Training Data Sources

We source all of our data from the following open-source TensorFlow datasets (Pot et al., 2019):

**Text** Since most of TensorFlow's text datasets are quite small, we concatenate the following five datasets into a single collection of 165GB: (i) *Wikipedia* (Wikimedia, 2023), the filtered UTF-8 encoded text from an XML dump from 2023-06-01,

containing all languages but predominantly English and western languages (113.9GB); (ii) *PG-19* (Rae et al., 2020), books from the Project Gutenberg, also encoded in UTF-8 (9.4GB); (iii) *Big Patent* (Sharma et al., 2019), a dataset of patents in English (30.2GB); (iv) *Scientific Papers* (Cohan et al., 2018), from arXiv and PubMed, containing the raw text including the LaTeX code (8.1GB); and (v) *Natural Instructions* (Mishra et al., 2022; Wang et al., 2022), tasks formulated in English covering different domains and lanugages (4.1GB).

**Image** We collect a subset of 165GB of the ImageNet dataset (Russakovsky et al., 2015), uniformly sampled across the 1000 classes, which contains 14 197 122 annotated images (of varying resolutions) from the WordNet hierarchy. We decode the images into RGB arrays (three `uint8` channels), flatten them, and concatenate them into a byte stream of flattened images. As a consequence, we ignore image boundaries when sampling from this data source (i.e., sequences are not guaranteed to start or end at the start or end of an image).

**Audio** We create a subset of 165GB from the Common Voice dataset (Ardila et al., 2020), a multilingual dataset of voice recordings. We downsample the dataset from 48 kHz to 16 kHz and encode the waveform as `int16`, i.e., with two bytes per sample. As for images, we concatenate all individual audio samples into a single byte stream. Accordingly, there is no guarantee that a sequence sampled from our dataset starts or ends at the beginning of a recording.

### B.2. Out-of-Distribution Evaluation Data Sources

We source all of our data from the following open-source TensorFlow datasets (Pot et al., 2019):

**Text** We use a 1GB subset of *Reddit* (Völske et al., 2017), which contains 3.8 million Reddit posts encoded in UTF-8.

**Images** We create a 1GB subset of the CelebA HQ dataset (Liu et al., 2015) with a resolution of $512 \times 512$. We process the images in the same way as for our image training set, i.e., flattening and concatenation, and we subsample uniformly across classes of CelebA.

**Audio** We use 1GB from the *LibriSpeech* (Panayotov et al., 2015) dataset, which contains roughly 1000 hours of English speech data derived from audiobooks that have been segmented and aligned in the LibriVox project. The data is already in 16kHz (with a sample size of 2 bytes), and we simply concatenate samples into a single byte stream.

**Multimodal Evaluations** For our evaluations on multimodal data, we use the unimodal evaluations on 1GB of data as described above and average the results accordingly (both for our models but also all standard compression algorithms, and Bellard's online adaptive transformer), either over two or three evaluations depending on the evaluation mixture composition.

### B.3. Sweeps

**Model Size vs. Dataset Size** The experiment to investigate the impact of training dataset- and model size, with results shown in Fig. 4, used the following model parameters. Dataset sizes were $20\%$, $40\%$, $60\%$, $80\%$, and $100\%$ of the full 165GB for each training set mixture (uni- and multimodal). All models used a context size of 4096, 8 attention heads per layer, a widening factor of 4 and the number of layers was either 2, 4, 6, 8, or 10. Models were trained with a batch size of 32. The learning rate was $1 \times 10^{-4}$, and a sinusoid positional encoding was used.

**Model Size vs. Context Size** Fig. 5 in the main paper shows the relationship between context length and model size. For this experiment we performed a large-scale sweep with the goal of covering a good range of training FLOPS budget with models that make various trade-offs between model size and context length (given the same model size, compute demand increases with increasing context length). The main question was whether there is a qualitatively similar relationship across parameters, and whether there is a clear sweet spot — see the main paper for results and discussion. For our sweep we used the same model parameters as in the previous paragraph (the training data size was always at $100\%$) and sweep over the following four context sizes (with training batch size in brackets): $[1024\ (128), 2048\ (64), 4096\ (32), 8192\ (16)]$. For each context size we train five models (XS, S, M, L, and XL) on all three unimodal datasets, respectively. Each model has a different combination of embedding dimension and number of layers for each different context size. The XS models have embedding dimensions $[112, 96, 80, 64]$ and numbers of layers $[11, 7, 5, 3]$ for the different context sizes respectively (i.e., wider and deeper models for shorter contexts and more narrow and more shallow models for long context size).

Table A2: Best compression ratios for each compressor. This table shows the same results as Fig. 2 but as precise numerical values to facilitate detailed comparison.

| Evaluation Modality | Out-of-Distribution Compression Ratio | | | | | | | |
| --- | --- | --- | --- | --- | --- | --- | --- | --- |
| | **Ours** | **Bellard** | **gzip** | **LZMA2** | **FLAC** | **PNG** | **JPEG 2000** | **JPEG-XL** |
| Audio | **0.487** | 0.509 | 0.813 | 0.699 | 0.538 | - | - | - |
| Image | 0.285 | 0.281 | 0.698 | 0.545 | - | 0.426 | 0.390 | **0.260** |
| Text | 0.217 | **0.204** | 0.394 | 0.286 | - | - | - | - |
| Audio + Image | **0.393** | 0.395 | 0.756 | 0.622 | - | - | - | - |
| Audio + Text | 0.362 | **0.357** | 0.604 | 0.493 | - | - | - | - |
| Image + Text | 0.270 | **0.243** | 0.546 | 0.415 | - | - | - | - |
| Audio + Image + Text | 0.349 | **0.331** | 0.635 | 0.510 | - | - | - | - |

Table A3: Running times to compress 1GB of data for all compressors used in our study. We use the best model per modality, which have different sizes and thus different running times.

| Evaluation Modality | Running Times [s] | | | | | | | |
| --- | --- | --- | --- | --- | --- | --- | --- | --- |
| | **Ours** | **Bellard** | **gzip** | **LZMA2** | **FLAC** | **PNG** | **JPEG 2000** | **JPEG-XL** |
| Audio | 305 609 | 101 178 | 55 | 524 | 169 | - | - | - |
| Image | 222 065 | 103 391 | 47 | 436 | 174 | 495 | 99 | 871 |
| Text | 452 355 | 100 657 | 102 | 881 | 184 | - | - | - |

The S models have embedding dimensions $[192, 160, 112, 96]$ and numbers of layers $[10, 8, 6, 4]$. The M models have embedding dimensions $[224, 192, 144, 112]$ and numbers of layers $[12, 9, 7, 5]$. The L models have embedding dimensions $[272, 240, 176, 144]$ and numbers of layers $[13, 10, 8, 5]$. The XL models have embedding dimensions $[320, 304, 240, 160]$ and numbers of layers $[12, 9, 7, 6]$. The main goal with these settings is to create families of models that have roughly the same demand in terms of FLOPS (iso-FLOPS) but very different trade-offs in terms of model- and context size.

### B.4. Computational Resources

We trained every model on 16 NVIDIA A100 GPUs from our internal cluster. We trained 315 models in total, yielding a computational footprint of 5040 A100s. We ran Bellard's code on an NVIDIA GeForce RTX 4090 GPU with a 24-core Intel i9-13900KF CPU @ 3Ghz.

## C. Additional Results

### C.1. Compression Ratios

Table A2 shows the optimal compression ratios that each of the compressors achieve on all of the different evaluation modalities (note that all evaluations are on out-of-distribution data). The same values as shown in Fig. 2 in the main paper and given here as precise numerical values for completeness.

### C.2. Running Times

Computing the FLOPS for standard compressors (e.g., gzip or PNG) is much more involved than for neural networks and requires intricate knowledge of the algorithm, which can consist of more than 100K lines of code (standard compression algorithms have also been highly optimized and run on CPU, whereas the neural compressors need a GPU). Therefore, we instead compare the wall-clock running times in seconds when compressing 1GB of data from each of the three modalities for our models, Bellard's online adaptive transformer (Bellard, 2021), and the standard compression algorithms used in our work. As Table A3 clearly shows, our models and Bellard's model are orders of magnitudes slower (let alone the increased computational demand and GPU requirements). Running times for our models differ, because we pick the best model per modality, which are models of different sizes.

Table A4: Compression ratios for model parameters. We losslessly compress the trained model parameters with standard compressors. For each modality we choose the best-performing model. As is shown, the maximal compression is 11%, which would affect the overall compression ratio on the corresponding evaluation data only very marginally.

| | Model Parameter Compression Ratio | |
| Evaluation Modality | gzip | LZMA2 |
| --- | --- | --- |
| Audio | 0.93 | 0.90 |
| Image | 0.93 | 0.90 |
| Text | 0.92 | 0.89 |

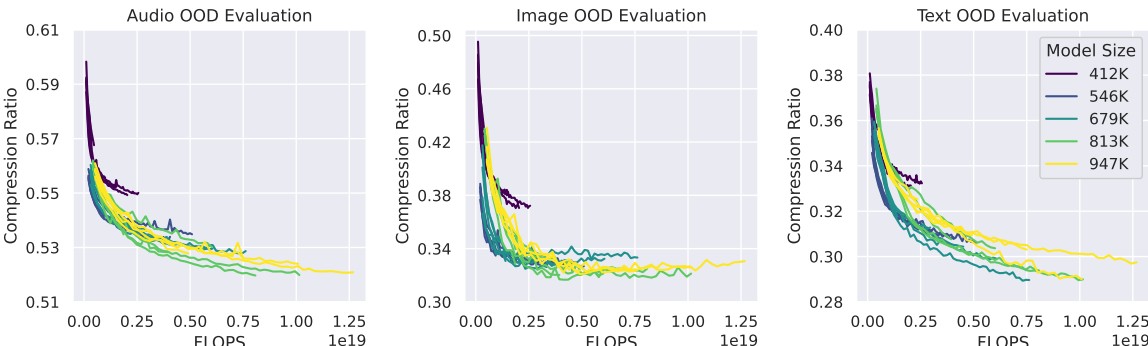

Figure A1: Similar to Fig. 4 in the main paper, but here the models are trained on a uniform mixture over all three modalities (55GB per modality). The plot shows compression performance evaluated on the unimodal datasets as training progresses for various model- and training set sizes (models are different colors, each line is a different training set size of either 20%, 40%, 60%, 80%, and 100%). We always train for 2 epochs, regardless of dataset size, i.e., smaller datasets require fewer FLOPS. In contrast to Fig. 4, where models are trained on unimodal data, we observe no overfitting, e.g., on images, even for the largest models tested. However, the compression ratios are slightly worse than for unimodal training, which is in line with our other expriments that show small losses when training on multimodal data.

## C.3. Compressing Model Parameters

Throughout our paper we report compression rates that take uncompressed model parameters into account. As discussed in the main paper, compression ratios could be improved by also compressing model parameters. However, as Table A4 shows, naively compressing model parameters with a lossless compressor does not lead to much compression, which would translate into very marginal gains on the overall compression ratio. While it is possible to investigate more sophisticated compression schemes, in particular lossy compression of network weights (though this opens the problem of having to solve a trade-off between increasing weight compression and maintaining compression performance), this is beyond the scope of our paper. Accordingly, our compression rates can be understood as (somewhat) conservative estimates that give (in our case fairly tight) upper bounds on compression performance. Compressing network weights to achieve competitive compression ratios would be of greater significance in a regime where models are significantly larger than ours (but the evaluation data stays roughly at the same size).

## C.4. Scaling Analysis for Multimodal Training

Fig. A1 shows the results of simultaneously scaling dataset- and model size across training. In contrast to the similar Fig. 4 in the main paper, where models were trained on unimodal data, Fig. A1 shows models trained on multimodal data (i.e., the uniform mixture across all three modalities, with 55GB per modality). The multimodal training mixture acts as a regularizer, which can clearly be seen by the lack of overfitting of the largest models on images. Compare this against the unimodal training results in Fig. 4 where overfitting can be observed. In line with our other main results in Fig. 2 and Fig. 3, the overall compression ratios are slightly worse for the models trained on multimodal data compared to unimodal training.

