# OpenReview forum: "Compression via Pre-trained Transformers: A Study on Byte-Level Multimodal Data"
_ICML.cc/2025/Conference — ICML 2025 poster_

### Official Review · Reviewer_YU4t · 2025-03-10

**Overall Recommendation:** 3

**Summary:**

This paper investigates the compression efficiency of foundation models when accounting for parameter size. Through extensive experiments on 165GB of text, image, and audio data, the authors demonstrate that relatively small transformer models (millions of parameters) can outperform standard compression algorithms like gzip and even domain-specific compressors like FLAC—achieving a 0.49 compression ratio on out-of-distribution audio data compared to FLAC's 0.54. While these smaller models can handle multiple trained modalities effectively, they show limited transfer capabilities to unseen modalities unlike their larger counterparts.

**Claims And Evidence:**

Although I appreciate the author's interesting findings that indicate a small transformer with massive training data can compress well to out-of-distribution data than general-purpose and even domain-specific compressors, the theoretical contribution is relatively weak for the ICML audience. Meanwhile, Deletang et al. actually have shown the performance of a transformer compressor trained from the scratch. Compared with Deletang et al., the main contributions of this paper lie on the usage of more multi-modality data and corresponding performance on cross-modality data, if I understand correctly. I still think the theoretical contributions are still weak to ICML audience.

**Essential References Not Discussed:**

N/A

**Experimental Designs Or Analyses:**

Indeed, there are some SOTA image lossless codecs are used for comparison. Please consider some representative SOTA image lossless codecs, such as JPEG-XL and WebP.

**Methods And Evaluation Criteria:**

I think the method is very easy to follow. Authors leverage a small transformer with multiple domains for training. The out-of-distribution compression performance is focused on.

**Other Comments Or Suggestions:**

Please see the weakness.

**Other Strengths And Weaknesses:**

Strengths:
- This paper is well-written and easy to follow

- The topic is interesting and the capacity of transformer-based model for lossless data compression is an important direction

- The experimental settings are comprehensive, and the authors choose multiple modalities for lossless compression comparison.

Weaknesses:
- Although I appreciate the author's interesting findings that indicate a small transformer with massive training data can compress well to out-of-distribution data than general-purpose and even domain-specific compressors, the theoretical contribution is relatively weak for the ICML audience. Meanwhile, Deletang et al. actually have shown the performance of a transformer compressor trained from the scratch. Compared with Deletang et al., the main contributions of this paper lie on the usage of more multi-modality data and corresponding performance on cross-modality data, if I understand correctly. I still think the theoretical contributions are still weak to ICML audience.

- I think it is necessary to compare with state-of-the-art lossless image compressor such as JPEG-XL.

- How the authors to process the RGB images for lossless compression? Deletang et al.  give the scheme of grayscale image. It is very interesting to claim this detail.

- Overall, I think this paper needs to further justify the contribution compared to previous literature. I am happy to increase my score upon nice answers.

**Questions For Authors:**

Please see the weakness.

**Relation To Broader Scientific Literature:**

N/A

**Theoretical Claims:**

This paper lacks theoretical claims and main results are derived from empirical experiments.

---

> ### Author Rebuttal · Authors · 2025-03-31
>
> We thank the reviewer for their careful assessment and constructive feedback. We are pleased that they think that our `topic is interesting and an important direction`, that our `experimental settings are comprehensive`, and that our `paper is well-written`.
>
>
> **Could you also compare to state-of-the-art lossless image codecs such as JPEG-XL?**
>
> Yes, thank you for the suggestion! We evaluated JPEG-XL on our image data and obtained a compression ratio of 0.26, which, as the reviewer correctly predicted, is slightly better than our model (and Bellard’s):
>
> | Evaluation Modality | Ours | Bellard | gzip | LZMA2 | PNG | JPEG 2000 | JPEG XL |
> | ------- | ------- | ------- | ------- | ------- | ------- | ------- | ------- |
> | Image | 0.285 | 0.281 | 0.698 | 0.545 | 0.426 | 0.390 | **0.260** |
>
> We have updated all the tables and figures with these results in our revised manuscript. Note that the point of our paper is not so much that neural networks can outperform any domain-specific compressor but that they can be on par with some of the most widely used general-purpose and domain-specific compressors.
>
>
> **I think that the theoretical contributions are too weak for the ICML audience**
>
> Our work is a comprehensive empirical study, which is in scope for and frequently gets published at ICML.
>
> The theory underlying our work (and Delétang et al. (2024)) is standard textbook material in information theory. Accordingly, Delétang et al. (2024), which was published at ICLR 2024, did not introduce this theory. Instead, they provided a concise review of that theory, as we did in Section 2.
>
>
> **What is the contribution compared to prior work?**
>
>  We investigated the *open* problem of whether *small* transformers *pre-trained* on large *multimodal* datasets can achieve competitive compression ratios across different modalities and whether there is *transfer to unseen modalities* (see Section 1), as observed in the large-scale model case. Prior work either considered large-scale, pre-trained models (Delétang et al., 2024) or models trained online (Bellard, 2021). As a result, we investigated a different regime that induces entirely different constraints on the optimal model architecture (in particular, its size) and for which it was a priori unclear whether competitive compression ratios would be achievable across different modalities.
>
> Accordingly, our contributions compared to the previous literature are (see also Table A1):
> * Small transformers can achieve competitive compression ratios on audio and image data (Delétang et al. (2024) only showed this for text data), even when comparing against domain-specific compressors, which the reviewer considers an `interesting finding`.
> * Small transformers can achieve competitive compression ratios across multiple domains, pointing towards the generality of relatively small models.
> * Unlike the results on large language models by Delétang et al. (2024), small transformers do not achieve transfer outside the training modalities. Moreover, unlike our work, Delétang et al. (2024) did not investigate in-modality out-of-distribution data (e.g., training on ImageNet and evaluating on CelebA) to show in-modality generalization.
>
>
> **How do you process RGB images for lossless compression?**
>
> As described in Appendix B.1, we use all three (RGB) channels, each represented as `uint8` (unlike Delétang et al. (2024), who compress grayscale images, i.e., use a lossy representation of the original image). We flatten the image and concatenate all flattened images to form our image byte stream from which we iteratively sample chunks where the chunk size depends on the model’s context size (typically only parts of a single image fit into our models’ context, and, occasionally, there may be parts of two different images in the context).

---

### Official Review · Reviewer_L7Gi · 2025-03-13

**Overall Recommendation:** 4

**Summary:**

The paper shows that small decoder-only transformers trained on *multimodal data* are effective data compressors. This happens when the modality of the data being compressed (text, audio, video) belongs to the training set. In this multimodal setup, the authors show that it is possible to achieve compression ratios that are competitive with state-of-the-art algorithms (on 1GB of data), even considering the transformer parameters.

**Claims And Evidence:**

The paper is well written the claims are supported by convincing evidence. See the experimental design section.

**Essential References Not Discussed:**

Essential references are discussed.

**Experimental Designs Or Analyses:**

A convincing and comprehensive set of experiments backs the paper.
The paper examines various effects influencing the compression rations, including the training set composition Fig.2, 3 model size Fig. 4. It explores how the context size and sliding window approaches affect the data compression and how the compression ratio is affected by the size of the evaluation set Fig. 7. Taken together these are a convincing set of experiment supporting the claims of the paper.

**Methods And Evaluation Criteria:**

The methods and evaluation criteria are appropriate and sound.

**Other Comments Or Suggestions:**

None; the paper is well written.

**Other Strengths And Weaknesses:**

The work is good and well-written, it is slightly incremental with respect to the previous studies on compression but addresses a relevant open question backed by a convincing and comprehensive set of experiments.

**Questions For Authors:**

I do not have any relevant concerns that could change my evaluation of this paper.

**Relation To Broader Scientific Literature:**

The relation to the broader scientific literature is thoroughly discussed in Sec. 6 and expanded in Sec. A of the appendix. The literature itself is well presented in Sec. 3.

**Theoretical Claims:**

No theoretical claims to discuss.

---

> ### Author Rebuttal · Authors · 2025-03-31
>
> We thank the reviewer for their thorough assessment of our work. We are pleased that they think that our `work addresses a relevant open question backed by a convincing set of experiments` with `methods and evaluation criteria that are appropriate and sound`, and that they consider our `paper well-written`.

---

### Official Review · Reviewer_Bivq · 2025-03-14

**Overall Recommendation:** 4

**Summary:**

This paper studies byte language models as a means to compress data on multimodal data (text, audio, images).  They show that by training small scale transformers data can be compressed better than with standard compression algorithms like gzip or domain-specific standards (JPEG). This is achieved without online training, i.e., the Transformers are not trained to compress the data they are evaluated on, unlike competing methods like Bellard.

The obvious disadvantage of this method is speed (much slower than standard compression). Thus the practical utility of the method is limited to very specific cases (where compression ratio has to be minimized, no matter the computational cost). Yet the authors are open about this, thus this is more of a feasibility study.

## update after rebuttal
I had initially given an "accept", conditional on the authors' response to clarify my questions (and concerns). The authors answered my questions, thus I retained my score. The work appears sound, novel and non-incremental, thus I am clearly in favor of acceptance.

**Claims And Evidence:**

There are four claims in the introduction which are all supported well by evidence:
(1) Empirical study on compression of small Transformers
(2) transformers are better at compressing than general-purpose compressors
(3) Multimodal boosts compression on multimodal data
(4) Poor transfer to unseen modalities

However, there is one implicit claim where I see zero support: Can you demonstrate that the model is actually doing lossless compression and the original sequence can be recovered exactly? How do you ensure that compression is lossless? Is this by construction? E.g., if you ran an inference before even starting the training, would the "compressed" file still be lossless, just exactly as big (or bigger) than the source bytestream? For a text-only model, I fail to see how it can losslessly compress 1GB of image.

**Essential References Not Discussed:**

None

**Experimental Designs Or Analyses:**

Apologies for my lack of knowledge on compression but I would like to see more details on the training/inference workflow for the model. Maybe a figure could help?
Training: If training is done with pure CE objective on next byte prediction, how is the compressed output extracted from the model? Training with standard next token (aka byte) prediction would yield a byte sequence of identical length to the input bytestream,.
Inference: If you compress 1GB of data, with a context size of 4096 and no overlap, then, in practice you sequentially run 1M/4096 forward steps. How many bytes are you predicting in each of those steps?


It's problematic to (1) cherry-pick and report the best performance over hyperparameter sweeps on the validation data (it seems that you did not do a train/validation/test split) and (2) report single-run results without any indication of confidence intervals. If you did only either (1) or (2) I wouldn’t raise this, but in combination, I find the results a bit brittle. I acknowledge that (1) prior art didn’t even make this distinction between train/validation and (2) the computational costs for such methods are very high. For the rebuttal I would like to see performance of all models, rather than only the best ones

**Methods And Evaluation Criteria:**

The compression ratio metric makes sense, but it is the only metric reported throghout the paper. Results may be slightly richer if some additional metrics (like train/validation loss) were also shown

**Other Comments Or Suggestions:**

Minor things:
	- Introduction: "LLMs compress better when parameter count is not considered" -- according to which metric are LLMs worse at compression when the count is considered? I only understood later from equation 3 that you include the size of the compressor by encoding model weights in half precision. Maybe you can be more explicit in the intro
	- L196 - missing "is"

**Other Strengths And Weaknesses:**

Strengths:
- Interesting findings that will impact the future of compression via LMs
- Paper is clearly written and relatively easy to follow
- Results are rich and multiple ablation studies were conducted

Weaknesses:
- The extraction of the bytes from the LM head is not clearly defined (see my comment above)

**Questions For Authors:**

None

**Relation To Broader Scientific Literature:**

Well related to broader literature in LM-compression space (Deletang, Bellard etc) and the byte-LM space (Megabyte, BLT etc). Out of curiosity, could the authors comment on the impact of larger context windows for compression algorithms? Megabyte has a context window up to 1M bytes, multiscale byte LMs (https://arxiv.org/pdf/2502.14553) can even go to 5M bytes -- would you expect that compression ratio can be increased significantly with a model that can fit the entire bytestream into the context window?

**Theoretical Claims:**

What is considered lossless compression? Are we talking about half, single, double precision? (I only saw that model weights are counted with half precision)

---

> ### Author Rebuttal · Authors · 2025-04-01
>
> We thank the reviewer for their thorough review and constructive feedback. We are pleased that they think our `interesting findings will impact the future of compression via LMs`, our `results are rich with multiple ablation studies`, and our `paper is clearly written`.
>
> **How do you ensure that your compression is lossless? Is this by construction?**
>
> Yes, it is lossless by construction as we use arithmetic coding (AC), a lossless compression algorithm that uses the next-symbol probabilities of a sequential predictor (i.e., our transformers) to compress (see Figure 1 in Delétang et al. (2024) and Hutter et al. (2024)). For any predictor, AC is optimal in terms of coding length, i.e., the compression performance only depends on the predictor. If the predictor is bad (e.g., an untrained network), the compressed size would be larger than the original — but the data would still be losslessly recoverable.
>
> To decompress, AC must use the same predictor, so we also have to communicate its parameters (we use half precision). The precision used to encode the parameters does not affect the compression's “losslessness”, only the performance.
>
> **How can a text-only model losslessly compress images?**
>
> As we convert all data into byte streams (no tokenization) and train our models on next-byte prediction, they can process any byte data. Images are simply byte streams with different statistics and can also be fed to models trained only on text bytes. If the model has learned general patterns, it will also predict the image byte stream well (and losslessly compress it well via AC).
>
> **What is considered lossless compression (half, single, double precision)?**
>
> Our compression is lossless by construction, and the numerical precision depends entirely on the AC, which can be arbitrarily high (at the expense of compression performance). We use the data’s precision, i.e., single precision (we compress bytes). The underlying data may be encoded at higher precision (e.g., `float32` images), which is losslessly recoverable from our byte streams.
>
> **Do you have a train/validation/test split, or did you cherry-pick the best models?**
>
> Our setup differs slightly from the standard train/validation/test splits as we always evaluate OOD data (e.g., train on ImageNet, evaluate on CelebA) to mimic the standard compressor setup. As the reviewer states, Figure 2 is a feasibility result, which is why we show the best performance. Figures 3 to 6 show *all* the models (the full hyperparameter sweeps), as the reviewer requested.
>
> **How are the bytes extracted from the model’s head during training and inference?**
>
> AC directly uses the model’s predictions over tokens (i.e., the logits) to encode/decode data (see Figure 1 in Delétang et al. (2024) for an overview). The model computes a distribution over the next byte for every input byte, and the AC then uses these predictions to losslessly compress the data. No separate head or extraction procedure is necessary.
>
> We train via standard log-loss minimization to perform next-byte prediction. At inference time, we perform standard autoregressive evaluation (using teacher forcing).
>
> We have added the above clarifications to our revised manuscript.
>
> **Why are LLMs worse at compression when the parameter count is considered?**
>
> LLMs are viable compressors depending on whether their parameter count is factored into the *compression ratio* (i.e., adjusted vs. unadjusted). Imagine that Alice wants to compress and send data to Bob, who wants to decompress it. If Alice trains a neural network to compress the data but does not communicate the model’s weights, Bob cannot decode the data. Thus, the parameter count needs to be factored into the compression ratio. We believe that the unadjusted compression ratio is not well suited to evaluate *pre-trained* neural networks as compressors.
>
> We have reformulated Section 1 to make this more explicit.
>
> **What is the context window's impact on compression performance?**
>
> We investigated this question in Figure 5. For text, increasing the model size is more beneficial than increasing the context length (short-term dependencies are most important). In contrast, large context sizes are generally beneficial for images (models with larger contexts can process larger fractions of an image). For audio, the relationship between model and context size is more complex.
>
> We expect MegaByte (Yu et al., 2023) or multiscale byte language models (Egli et al., 2025) to improve performance on some but not all domains.
>
> **Why do you only report the compression ratio?**
>
> As we train our models for lossless compression via AC (which is equivalent to log-loss minimization), the compression ratio is the only relevant metric (apart from more “practical” metrics such as running time, which we report in Table A3). The compression ratio and the log-loss are proportional (the additive factor is the model size) and can be recovered from each other.
>
> **L196 misses "is".**
>
> Fixed, thanks!

---

> > ### Comment · Reviewer_Bivq · 2025-04-02
> >
> > I'd like to thank the authors for the clear explanations and congratulate to the great work!
> >
> > Last note: I stumbled over the point that all evaluations were done with teacher forcing as this is very different from standard LM evaluation. I see why this is not problematic here because you are always just modeling the "prompt" and never do any free generation, but I still think that this should be clarified in the final paper

---

> > > ### Author Response · Authors · 2025-04-02
> > >
> > > Yes, that's a very good point, and we will clarify this in the revised manuscript. Thank you for all your help in improving our submission!

---

### Official Review · Reviewer_5Jx7 · 2025-03-15

**Overall Recommendation:** 3

**Summary:**

1. This paper empirically examines the effectiveness of small pre-trained transformers (with millions of parameters) as multimodal data compressors for text, images, and audio.
2. Trained on 165GB of data per modality, these models achieve compression ratios that surpass both general-purpose and domain-specific compression algorithms while remaining competitive with state-of-the-art online adaptive transformers. However, despite their strong performance on in-domain data, traditional compressors outperform them on out-of-distribution data.
3. The study also investigates the effects of multimodal training, revealing limited cross-modal transfer in small transformers—contrasting with trends observed in large-scale models.

**Claims And Evidence:**

yes

**Essential References Not Discussed:**

no

**Experimental Designs Or Analyses:**

Yes, The authors compare to various standard compressors, both general-purpose, and domain-specific for audio data and PNG and lossless JPEG2000  for images. They also compare to the online transformer with the default v3.3 parameters, which is currently SOTA on the Large Text Compression Benchmark(LTCB).

**Methods And Evaluation Criteria:**

Yes, The evaluation is thorough, leveraging appropriate benchmark datasets and comparing performance against both domain-specific and general-purpose compression algorithms.

**Other Comments Or Suggestions:**

no

**Other Strengths And Weaknesses:**

Strengths
1.The writing is clear and well-structured, making the study easy to follow.
2. This paper presents a comprehensive empirical study on the compression performance of small transformer models, particularly for multi-modal data.
3.Key findings:
•	Small pre-trained transformers can achieve compression rates comparable to both domain-specific and general-purpose algorithms.
•	Even with multimodal training, small transformers do not develop a universal compression ability that enables strong performance on unseen modalities.

Weaknesses:
1.	Limited novelty: The study appears to be an extension of [1], applying the pre-training of small transformers for text compression to a multimodal setting with additional ablation experiments. However, it does not seem to derive significantly more valuable insights or practical applications beyond this extension.
2.	Intuitive conclusions: The conclusions of the paper seem intuitive and straightforward. While there may not be a prior work that explicitly derives these specific findings, existing literature on the multimodal generalization capabilities (in this case, their compression abilities) of both large and small models  already points toward similar conclusions. The paper could be strengthened by going beyond empirical observations and offering optimization strategies—such as transfer learning—to improve small transformers' performance on unseen modalities, making the findings more actionable and impactful.
[1] Delétang G, Ruoss A, Duquenne P A, et al. Language modeling is compression[J]. arXiv preprint arXiv:2309.10668, 2023.

**Questions For Authors:**

Please refer to weakness part.

**Relation To Broader Scientific Literature:**

This is my main concern: the conclusions of the paper seem intuitive and straightforward. While there may not be prior work that explicitly derives these specific findings, existing literature on the multimodal generalization capabilities (in this case, compression abilities) of both large and small models already points toward similar conclusions.

**Theoretical Claims:**

Key findings:
•	Small pre-trained transformers can achieve compression rates comparable to both domain-specific and general-purpose algorithms.
•	Even with multimodal training, small transformers do not develop a universal compression ability that enables strong performance on unseen modalities.
The authors present two key findings (as highlighted in the summary). Building on prior work [1][2], they conduct a comprehensive study on multi-modal training and perform extensive ablations over context size, dataset composition, and model size, ultimately deriving conclusions specific to small transformers.
[1] Izacard G, Joulin A, Grave E. Lossless Data Compression with Transformer[J]. 2019.
[2] Yu L, Simig D, Flaherty C, et al. Megabyte: Predicting million-byte sequences with multiscale transformers[J]. Advances in Neural Information Processing Systems, 2023, 36: 78808-78823.

---

> ### Author Rebuttal · Authors · 2025-03-31
>
> We thank the reviewer for their positive feedback and insightful comments. We are pleased that they think that our `paper presents a comprehensive empirical study`, which `leverages appropriate benchmarks`, and that our `writing is clear and well-structured`.
>
>
> **What is the novelty over Delétang et al. (2024)?**
>
> While our work is indeed inspired by Delétang et al. (2024), we believe that the unadjusted compression ratio that they primarily use to evaluate their models is not well suited to evaluate pre-trained neural networks as compressors since it does not capture the entire output of the compressor. The adjusted compression ratio, which is the one we investigate, induces an entirely different regime where large-scale models are no longer viable. Note that Delétang et al. (2024) are aware of this and, therefore, also report the adjusted compression ratio for their models, with the difference that the large language models (LLMs) they evaluate are not competitive in this regime.
>
> Moreover, Delétang et al. (2024) focus on off-the-shelf LLMs and only perform a single experiment where they train small transformers on enwik8 (100MB of text data) to obtain strong compression performance on enwik9. In contrast, we perform a `comprehensive empirical study` (in the reviewer’s words) in the multimodal setting with much larger datasets (165GB), where we not only show that small-scale transformers can achieve compression ratios on audio and image data that are on par with general-purpose and domain-specific compressors but also investigate the cross-modality transfer (in addition to a variety of ablations). We also improve the evaluation by always evaluating on out-of-distribution data (even within the same modality) to ensure a fair comparison with standard compression algorithms (overfitting to the training distribution is no longer possible).
>
> Accordingly, our contributions compared to the previous literature are (see also Table A1):
> * Small transformers can achieve competitive compression ratios on audio and image data (Delétang et al. (2024) only showed this for text data), even when comparing against domain-specific compressors.
> * Small transformers can achieve competitive compression ratios across multiple domains, pointing towards the generality of relatively small models.
> * Unlike the results on LLMs by Delétang et al. (2024), small transformers do not achieve transfer outside the training modalities. Moreover, unlike our work, Delétang et al. (2024) did not investigate in-modality out-of-distribution data (e.g., training on ImageNet and evaluating on CelebA) to show in-modality generalization.
>
>
> **The conclusions of the paper seem intuitive and straightforward.**
>
> While we agree that our results may not challenge the expectations and intuitions of experts in the field who know the literature well, we present novel quantitative results, which are significant and address an important *open* problem, i.e., whether small transformers pre-trained on large (multimodal) datasets can achieve competitive compression ratios across different modalities and whether there is transfer to unseen modalities, as observed in the large-scale model case (see Section 1). A priori, it was unclear whether we would attain similar conclusions to existing literature on multimodal generalization of small and large models since the compression viewpoint introduces an entirely different set of constraints on the models (most importantly with respect to their size). Finally, even some of our qualitative findings, e.g., the lack of out-of-modality transfer even for our largest models that we trained on multiple different modalities, were somewhat surprising to us.
>
> We agree that investigating more sophisticated training strategies to improve cross-modal transfer, e.g., via transfer learning, presents an interesting direction for future work and have mentioned this in the updated version of our manuscript.

---

> > ### Comment · Reviewer_5Jx7 · 2025-04-08
> >
> > Thank you for the response. However, I still have concerns about whether the overall design of the paper and the conclusions are intuitive and straightforward. Although the authors provide systematic experiments to support their approach, further optimization is still lacking. Therefore, I will retain my original score.

---

### Decision · Program_Chairs · 2025-05-01

**Decision:**

Accept (poster)

**Comment:**

Novel work on studying byte-level multimodal model. The work is motivate,  and the experiments are solid and convincing. Reviewers concerns on the incremental contribution, results / conclusions are intuitive and straightforward, and lack of theoretical contribution.